# Endometriosis of the Canal of Nuck: A Systematic Review of the Literature

**DOI:** 10.3390/diagnostics11010003

**Published:** 2020-12-22

**Authors:** Anastasia Prodromidou, Anastasios Pandraklakis, Alexandros Rodolakis, Nikolaos Thomakos

**Affiliations:** 1st Department of Obstetrics and Gynecology, Alexandra Hospital, National and Kapodistrian University of Athens, 11528 Athens, Greece; tasospandraklakis@hotmail.com (A.P.); rodolaki@hol.gr (A.R.); thomakir@hotmail.com (N.T.)

**Keywords:** nuck cyst, nuck endometriosis, canal of nuck

## Abstract

Endometriosis is a common benign gynecological condition defined as the presence of endometrial tissue in tissues outside the uterine cavity. Apart from the common sites of endometriosis, rare sites other have also been reported including the liver, the thoracic cavity, the muscles, nerves, and more rarely in a patent Nuck canal. We aim to evaluate the clinical presentation, diagnostic features, and management of the Nuck endometriosis. A meticulous search of three electronic databases was performed until May 2020 for articles reporting cases of Nuck endometriosis. A total of 36 patients from 20 studies were analyzed. Median age of patients was 36 years with 33 women being of reproductive age. A right-sided lesion was identified in 30 cases (83.3%), while all patients suffer from a groin mass with cyclic pain in a proportion of 22%. All the patients finally underwent surgery for investigation of the lesion and fixation of the defect. Five cases of malignancy were detected at final pathology. All of them were alive with a median reported overall survival of 37 months. Nuck endometriosis should be included in the differential diagnosis of female patients with groin swelling. An evaluation by a gynecologist is important when endometriosis is suspected.

## 1. Introduction

Endometriosis is a common benign gynecological condition, which is characterized as the presence of endometrial tissue, containing glands and stroma, in tissues outside the uterine cavity [1]. The disease affects approximately 10–15% of women of reproductive age [1]. Although endometriosis can be asymptomatic, in many cases, it presents with pelvic pain, infertility, or incidentally at histological examination of an ovarian cyst [2]. Due to the close relationship of endometriosis with estrogen secretion, postmenopausal women are rarely affected [3]. The diagnosis of endometriosis can be suspected by evaluation of cyclic pain and symptoms and can be enhanced with imaging findings [4]. In recent years, the role of ultrasonography has gained significant popularity as diagnostic tool in the detection of pelvic endometriosis as well of less common extragenital lesions [4]. However, the definite diagnosis is only made by histologic confirmation of the excised specimens. The management of endometriosis is dictated by the severity of the symptoms as well as the wish of infertility correction [5]. Treatment options include medical, surgical, or combination of both therapeutic modalities [5]. To date, there is not enough evidence available comparing surgical with medical treatment and indicating the potential superiority of one of them. However, initial treatment with pharmaceutical regimens including analgesics (nonsteroidal anti-inflammatory drugs), GnRH analogs, and danazol seems effective in pain relief of a significant proportion of cases [6]. Surgical treatment with lesion resection and/or electrocauterization, nerve excision, or hysterectomy and oophorectomy through laparoscopy or laparotomy could be applied in cases of failure of the conservative treatment modalities [6]. The incidence of malignant transformation is low, but it is supported that if ovarian endometriomas is left untreated, there is a high probability of malignant transformation into endometrioid and clear cell carcinomas, the incidence of which reportedly ranges 0.6–1% [7,8]. There are also a few cases of malignant transformation of endometriosis into endometrial stromal sarcoma [9]. The most common locations of endometriotic lesions are usually the ovaries, the utero-sacral ligaments, the pouch of Douglas, and the bladder [10]. Nevertheless, there are also uncommon intra- and extrapelvic sites of detection of such as in the liver, in the thoracic cavity, in the muscles, and nerves [11,12]. 

The canal of Nuck (NC) was named after Anton Nuck in 1691 and represents the anatomical defect when a parietal peritoneal pouch follows the gubernaculum during its development [13]. The lower part of the gubernaculum becomes the round uterine ligament, which inserts the abdominal wall and runs through the inguinal canal along with the processus vaginalis [14]. The NC in the female system corresponds to the processus vaginalis in males. Normally the protrusion of parietal peritoneum obliterates during the nineth month of fetal life [15], but in cases of incomplete or failure to obliterate, it may manifest as a hydrocele, an indirect hernia or entrapment of various pelvic organs [16]. The pathology of the inguinal region includes a variety of diseases and arises from the different groin structures, complicating the differential diagnosis. Despite the fact that the pathology of the NC is not so widespread among either general surgeons or gynecologists due to the rarity of the NC diseases, they have been associated with severe morbidity and they need further investigation [16]. For the investigation of the pathology in the NC, the ultrasonography (US) has been claimed as a fast, inexpensive initial modality with significant diagnostic accuracy, while magnetic resonance imaging is performed in case of controversial US analysis [16]. Other diseases of the inguinal and vulva region such as Bartholin pathology, lymph node enlargement, abscesses, and soft tissue tumors are included in the differential diagnosis of the pathology in the canal of Nuck [16]. Endometriosis in the canal of Nuck is a quite rare surgical entity, which can be a diagnostic challenge in many cases. In worst scenario, the incarcerated endometriotic tissue can be transformed into a malignancy.

The aim of our systematic review was to accumulate the currently available literature on the cases of endometriosis in the NC with a particular focus in clinical presentation, diagnostic features, and management of the disease as well as the cases of malignant transformation of Nuck’s canal endometriosis.

## 2. Materials and Methods

### 2.1. Study Design

The Preferred Reporting Items for Systematic Reviews and Meta-Analyses (PRISMA) guidelines were followed for the design of this systematic review [17]. All appropriate observational studies, case reports, and case series that were written in English language and addressed cases of women who were diagnosed with endometriosis of the NC and underwent surgery were considered eligible for inclusion in the present review. The diagnosis could be either pre- or postoperative. Cases that were preoperatively considered as endometriosis but the final histology revealed other disease were excluded. Reviews and animal studies were excluded from analysis and tabulation. N.T., A.Pan., and A.Pr. independently performed a meticulous search of the literature, excluded overlaps, and tabulated the selected indices in structured forms.

### 2.2. Search Strategy and Data Collection

We systematically searched for articles published up to May 2020 using Medline (1966–2020), Scopus (2004–2020), and Google Scholar (2004–2020) databases along with the references of the articles, which were retrieved in full text. The search included the following key words: “Nuck canal,” “Nuck cyst,” “Nuck endometriosis,” “canal of Nuck,” and “endometriosis”. A minimum number of search keywords were utilized in an attempt to assess an eligible number that could be easily searched while simultaneously minimizing the potential loss of articles. Articles that fulfilled or were deemed to fulfil the inclusion criteria were retrieved and evaluated; all studies that described cases of women aged >18 years with no known developmental genital tract defect and were diagnosed and operated with cystic lesion along the NC with preoperative characteristics suspect for endometriosis or with lesions that were of endometriotic origin at final pathology were recruited. Additionally, cases of women with malignancy arose from endometriosis were also recruited. Other pathologies of the NC except the presence of endometriosis (i.e., hydrocele and nonendometriosis-related malignancy) were not considered eligible for inclusion to the present study. Accordingly, cases of endometriosis in the surrounding structures such as those referred as inguinal endometriosis were also not included. The PRISMA flow diagram schematically presents the stages of article selection (Figure 1).

Our search strategy included the MeSH terms:Nuck[All Fields] AND (“endometriosis”[MeSH Terms] OR “endometriosis”[All Fields])Nuck[All Fields] AND (“cysts”[MeSH Terms] OR “cysts”[All Fields] OR “cyst”[All Fields]) AND (“endometriosis”[MeSH Terms] OR “endometriosis”[All Fields]).

Data on patient characteristics included age and parity of the women. The main findings of the study site of the hydrocele along with the clinical appearance and diagnostic procedures (primary symptom and imaging) were appraised. Moreover, intraoperative and postoperative outcomes were as well evaluated, and type of surgical procedure and repair of the defect, length of hospital stay, histopathological outcomes of the excised specimens and follow-up data were enrolled if available. 

### 2.3. Quality Assessment

Case reports and case series are associated with elevated bias due to the nature of those types of studies [18]. Nonetheless, in the case where data on a certain condition is limited, evidence from those studies is considered of clinical importance. We evaluated the quality of the enrolled studies by adopting a quality assessment tool for case reports and case series proposed by Murad et al. More specifically, the methodological quality of the studies was assessed based on the criteria including the domains of ascertainment, causality, selection, and reporting. The sum of the scores derived from eight critical questions referred to the domains was used to evaluate the quality of each study as well as the reviewer’s judgement on the presence of the most important domains according to certain clinical case.

## 3. Results

### 3.1. Included and Excluded Studies

From the 702 records that were screened during the initial evaluation after excluding the duplicates, 23 were considered eligible after reading the abstract. Among them and after reading the full text, a total of 3 studies were excluded from the present review due to insufficient data [19,20,21]. More specifically, among them, one presented cases of external endometriosis among which one was characterized as NC endometriosis [19]. However, no separate outcomes for this case were provided by the authors and it was thus not included [19]. Respectively, no separate outcomes were identified in the study by Jimenez et al. for the one case of NC pathology among the cases of inguinal endometriosis that were reported and was excluded [20]. Finally, despite the fact that the study by Mazzeo et al. reported a case of vulvar endometriosis in a patient with previous surgery for Nuck pathology trying to investigate the contribution NC in the pathogenesis of the disease, no endometriosis in the canal was identified and the study was thus excluded [21].

A total of 20 studies (18 case reports and 2 case series), which comprised 36 patients who were diagnosed with an endometriotic lesion arose in the NC, were finally considered eligible for inclusion in the qualitative analysis of the present systematic review [22,23,24,25,26,27,28,29,30,31,32,33,34,35,36,37,38,39,40,41]. 

Table 1 depicts the main perioperative characteristics of each of the included studies with regards to primary symptoms in presentation, outcomes of clinical examination, as well as the type of performed surgical procedure and histopathology outcomes. 

Table 2 provides a cumulative report of the characteristics of the patients with endometriosis of the NC with malignant transformation. 

### 3.2. Quality Assessment

Based on the type of the included clinical cases, we considered the score of 5 points as the highest that could be assessed when excluding the three questions from the quality assessment tool that attributed to cases of adverse drug events. The scores of each study are presented in Table 1. A mean score of 3.6 (SD ± 1) was calculated, whereas the overall judgement on the quality of the recruited studies was that they were of moderate quality. 

### 3.3. Main Characteristics of Included Studies and Disease-Related Characteristics

Median age of patients included was 36 years (range: 20–64 years) while 33 (92%) of them were women of reproductive age. Four of the included women (36.4%) were nulliparous, while the remaining 7/11 (63.6%) women had one child or more. For the remaining 24 patients, data concerning parity were not available. A right-sided lesion was identified in 30 (83.3%), whereas in the remaining 6 (16.7%) patients, a left-sided one was noted. The enrolled patients, were admitted with a groin swelling or enlarging mass, which was either painful in 11/19 (58%) cases or painless in the remaining 8/19 (42%) cases. In 8 of the aforementioned cases, a cyclic progressively enlarged mass, which was associated with menstruation was recorded. Median maximum diameter of the lesion was 3 cm (range: 1–8 cm).

Information concerning investigation with imaging studies was available for 35 patients of whom 33 had and 2 did not have done it. More specifically, ultrasound (U/S), computed tomography (CT), magnetic resonance imaging (MRI), positron emission tomography (PET), or a combination of them were performed; 18 patients underwent a US imaging, a preoperative CT was performed in 10 patients, 12 patients underwent MRI, whereas in 2 cases, a PET scan was performed, among which one was indicative of the presence of hypermetabolic reactive inguinal lesion. In 27 of them, imaging was suspicious for presence of a cystic structure in the inguinal region which was compatible with pathology in the NC potentially indicative of Nuck hydrocele in most cases, while preoperative suspicion of the presence of endometriosis through imaging was made for 4 patients. Three patients underwent a preoperative histologically-based diagnostic procedure (biopsy or fine needle aspiration-FNA) [26,32,37]. In 2 of them, a preoperative percutaneous biopsy of the mass revealed endometrial tissue, while in the other patient, inguinal mass biopsy revealed adenocarcinoma [37]. 

Two out of 36 patients (5.5%) reported a history of pelvic endometriosis, whereas 4 had a history of previous obstetric or gynecological surgery among which only 1 was endometriosis related. The gynecologic procedures in the 3 other patients mainly included total abdominal hysterectomy and bilateral salpingo-oophorectomy due to uterine descensus, menometrorrhagia, and mucinous cystadenocarcinoma, not related to endometriosis [23,24,28].

### 3.4. Perioperative Outcomes

All the patients finally underwent surgery for investigation of the lesion and fixation of the defect. Data with regards to the type of surgery was available for all except one patient, as shown in Table 1, while excision of the mass was the main procedure performed. Two cases of laparoscopic approach for the investigation of the lesion have been reported by Jimenez et al. and Wang et al. [27,32]. Among them, Wang et al. performed a laparoscopic repair of the defect in the canal with simultaneous percutaneous excision of the endometriotic lesion, while in the study by Jimenez et al., a total laparoscopic excision and repair of the defect were applied [27,32]. Two patients also underwent radical hemivulvectomy due to malignant transformation of NC endometriosis [24,37]. Hernia repair along with the lesion excision was performed in 12 cases (1 laparoscopic and 11 open) among which 4 cases underwent repair with the use of mesh (Table 1). 

Histological examination confirmed the diagnosis of endometriosis and revealed the characteristic endometriotic features, which included the presence of endometrial stroma and glands. In cases in which immunochemistry was performed, ER and PR positivity was recognized as well as CK7-positive staining in the examined specimens [28,37]. Five cases of malignant transformation were histologically confirmed [22,23,24,28,37].

The length of hospital stay, as reported by 4 studies, ranged from 1 to 10 days, depending on the type and the approach. According to the data available in 9 of the included studies, recurrence rate of endometriosis was 1/10 (10%) in a follow-up period, which ranged from 2 to 37 months after surgery. The majority of the recurrences were noted in cases with detection of malignancy. 

### 3.5. Endometriosis-Related Malignancy

A total of 5 studies reported outcomes for 5 patients who were diagnosed with malignancy arising from endometriosis of the NC [22,23,24,28,37]. The median age of those patients was 50 years (range: 34–64 years). All except one of those patients had a right-sided mass. The type of surgical procedure for the initial management of the disease ranged from extensive local resection of the lesion with the surrounding structures to additional intraabdominal staging procedures including lymphadenectomies (Table 2). The histopathology of the resected specimens differed among the included cases and revealed 2 cases of endometrioid adenocarcinomas, one clear cell adenocarcinoma, a low-grade papillary adenocarcinoma, and a case of endometrial stromal sarcoma (ESS). FIGO stage was only available for the latter case of ESS referred as stage II [24]. Three patients developed disease recurrence, which was managed with respective resections as shown in Table 2, while in the case by Motooka et al., metachronous endometriosis-related malignancy of the endometrium and ovary were detected at 12 and 17 months, respectively, from initial diagnosis [37]. At follow-up, all patients were alive with no evidence of disease in 4 of them, while the patient reported by Mesko et al. was diagnosed with pulmonary metastasis but refused chemotherapy [23]. The median overall survival was 37 months (range: 12–96 months).

## 4. Discussion

The present study analyzed 36 cases of Nuck endometriosis from a total of 20 studies. Median patient age was 36 years while almost all of them were of reproductive age (92%). A right-side predominance was detected in more than 80% of the lesions of the NC, while all patients underwent surgical exploration of the mass with subsequent excisional procedures. Five cases of malignant tumors arising from the NC endometriosis were recorded. All of the aforementioned patients were alive in a median follow-up of 37 months.

The differential diagnosis of a groin lump or a subcutaneous mass of the inguinal region is challenging and may include a wide variety of entities such as hernia, lymph node enlargement, malignancy, endometriosis, Nuck hydrocele, lipomas, and abscess [42]. The presence of endometriosis in the inguinal canal is rare with an estimated prevalence of 0.3–0.6% of all endometriosis cases [37,43]. The age range of patients presented with groin lump compatible with endometriosis is 22–46 years [43]. If endometriosis in the inguinal region is suspected, it could be endometriosis within the inguinal hernia sac or endometriosis of the NC, the round ligament and the subcutaneous tissue [44]. A limited number of approximately 130 cases of groin endometriosis have been reported in the literature [45]. A painful lump with cyclical pain and enlargement is the most common presentation of endometriosis of the groin region [43]. The clinical presentation is similar irrespective of the tissue from which the endometriosis originates, disabling the differential diagnosis. To that end, preoperative imaging with US and/or MRI could be valuable, despite the fact that surgical exploration and histopathology could clearly set the origin and the final diagnosis of the disease [46].

Diagnosis of endometriosis in the NC is even more uncommon in patients with patent NC due to failure of the canal to close within the first year of life in females [47]. The aforementioned defect can result in hydrocele formation or herniation of intraabdominal surrounding tissues through the canal [47]. Under certain circumstances, which, however, remain unknown, the NC can be infiltrated by endometrial cells and results in the formation of endometriosis. A plethora of theories have been proposed to explain the presence of endometrial-like tissue outside the uterine cavity. One of them suggests that during retrograde menstruation, endometrial cells elude to the pelvic cavity via the fallopian tubes resulting in the formation of endometriotic implants throughout the pelvic structures [48]. This theory can also explain the local spread of endometrial tissue through the inguinal canal and the patent NC. Another theory advocates for the systemic lymphatic or hematogenous spread and aggregation of endometrial cells in distant organs as well as the deposition through the structures of the inguinal canal [43]. Finally, the third and most prevalent theory is that of coelomic metaplasia, which can construe the presence of endometriosis in patients with congenital (Mayer-Rokitansky-Kuester-Hauser syndrome) or iatrogenic lack of uterus (hysterectomized patients) [49]. This theory could be considered the potential mechanism of endometriosis formation in 3 of the included patients who underwent hysterectomy and had no previously known history of endometriosis. 

As proved in our study, endometriosis in the canal of Nuck is largely characterized by groin mass enlargement potentially associated with localized pain. It can be misdiagnosed as a typical inguinal hernia or Nuck hydrocele. In some cases, the gradual enlargement of the inguinal mass during menstruation and/or catamenial inguinal pain could be suggestive of the presence of endometriosis. Nonetheless, this was recognized only in a proportion of 8 patients in our series. A meticulous medical history from the patient with the groin mass suspected for endometriosis is critical for the preoperative diagnosis of the disease. In that setting, the patient should be thoroughly questioned about family history of endometriosis, history of any gynecological surgery with special consideration to previous caesarean section or myomectomy, and history of endometriosis and the respective treatment [50]. Patients with history of endometriosis are more likely to have concomitant disease of the inguinal canal through the round ligament. Furthermore, the characteristics of pain including frequency, duration, association with menstruation and location should be also thoroughly asked [50]. Similarly, a proper preoperative imaging is also of critical importance. Ultrasound has been widely utilized for the investigation of the pathology of the Nuck canal [16]. In the majority of the cases presented in this study, the U/S of the affected groin revealed a hypoechoic cystic lesion with internal echoic structures. In some cases, a CT, MRI, or even a PET-CT supplemented the U/S findings. High-resolution U/S is an inexpensive and reliable modality to set the suspicion of diagnosis of the Nuck endometriosis [47]. Nonetheless, the final diagnosis is based on the histology and immunochistochemistry. 

We noted a significant predominance of the right-sided endometriotic lesions of the Nuck canal. This finding is in accordance with the findings of previous studies in the pathology of the inguinal canal and the patent NC [41]. The most prevalent reason for this is the clockwise flow of the intraperitoneal fluid, which under the effect of gravity, promotes the prolonged stay of endometriotic cells in the right side and is further enhanced by the protective role of the sigmoid colon on the left side [43,44]. Another less common theory identifies the presence of a lymphatic pathway of atypical lymph nodes from the peritoneal cavity to the right groin which could potentially transfers endometriotic cells to the right-sided groin [43]. However, this does not apply in the case of right sided predominance of pelvic endometriosis [43].

As shown in our study, surgical resection with simultaneous repair of the defect of the canal in the majority of cases is the optimal treatment option for patients with endometriosis of the NC. Similarly to the case of Nuck hydrocele, the gold standard treatment remains excision of the cystic structure (laparoscopic or open) and closure of the internal inguinal defect with or without mesh [47]. Additionally, in case of extension of the disease to the labia majora, a concomitant repair of the vulva may be necessary including vulvectomy or clitorectomy procedures depending on the extent of the disease [51]. As shown in Table 1, the type of surgical procedure varies from transabdominal laparoscopic or open approach to excisional biopsy of the tumor through an inguinal approach. The approach is largely determined by the clinician’s estimation of the patient’s medical history, physical examination, symptomatology, laboratory testing, and preoperative imaging. The surgeon should also be aware so as to avoid dissemination of endometriotic cells to the surrounding tissues during surgical manipulations in order to reduce recurrences [43]. Additionally, the postoperative administration of hormonal suppression therapy is a matter of controversy. A variety of hormone suppression regimens including combined oral contraceptives, androgens, progestins, GnRH analogs, and levonorgestrel intrauterine device have been proposed for the decrease in disease recurrence after surgery for endometriosis [52]. According to a recent systematic review and meta-analysis, the risk of endometriosis recurrence was significantly reduced in patients who postoperatively received any kind of hormonal suppression compared to those who received placebo or no therapy (1766 patients, data from 14 studies RR 0.41, 95% CI 0.26 to 0.65, *p* < 0.01) [52]. The same was also observed in case of postoperative pain, which was significantly lower in the hormonal suppression group (652 patients, data from 7 studies, SMD −0.49, 95% CI: −0.91 to −0.07, *p* < 0.01) [52]. Nonetheless, due to the limited cases of endometriosis related to groin, there is no accurate data on the proportion of patients who received postoperative hormone therapy as well as on the respective recurrence rates.

Given the rarity of the presence of a patent NC, the detection of malignancy is an even rarer finding. Only a limited number of cases with malignant tumors arising from Nuck endometriosis have been reported in the literature. We herein analyzed a total of 5 cases with malignant disease. To date, no consensus has been reached with regards to the optimal management of Nuck endometriosis-related malignancy. Based on the available cases, complete surgical resection is the preferred treatment of the primary lesion. Among the 3 patients who developed recurrence, one had local recurrence, one had distant lung metastasis, while the remaining patient had initially local recurrence and developed lung metastasis 2 years after the first surgery. 

The present review is, to the best of our knowledge, the first in the literature, which presents a cumulative report of cases of endometriosis developed in the NC. A meticulous search of the literature eliminated the risk of potential loss of articles. However, there are some limitations that need to be addressed before reaching to safe conclusions. First of all, the estimation of the exact prevalence of Nuck endometriosis could not be precisely achieved, while robust conclusions concerning the pathophysiologic pathway of Nuck endometriosis formation, the clinical appearance of the disease, as well as the optimal treatment could not be reached. The currently available outcomes are limited to case reports and small cases series of moderate quality, which precludes further research due to the rare entity. Furthermore, there is significant heterogeneity among the included studies and some parameters were omitted by some of them which consists another critical factor to aggregate the currently available knowledge in the field. Finally, no sufficient information was available by the recruited studies with regards to the perioperative use of hormonal replacement therapy. More specifically, in the study by Uno et al., the preoperative hormonal therapy was prescribed aimed to reduce the size of the lesion without, however, any significant impact, whereas Wang et al. postoperatively prescribed a 3-month gonadotropin-releasing hormone agonist [27,35]. 

## 5. Conclusions

Despite its rarity, the Nuck endometriosis is an existing entity, which should not be neglected when investigating the pathology of the NC. The presence of endometriosis in the NC should be considered in the differential diagnosis of groin lump in female patients. Due to the fact that the pathology of the groin is mostly investigated by general surgeons, they should thoroughly ask for patients’ previous medical history and complaints with regards to the disease and consult a gynecologist when they suspect the presence of endometriosis. Additionally, the role of imaging and preoperative designation of the diagnosis should be highlighted so as for the cases of the presence of malignancy to appropriately design the management of the disease.

## Figures and Tables

**Figure 1 diagnostics-11-00003-f001:**
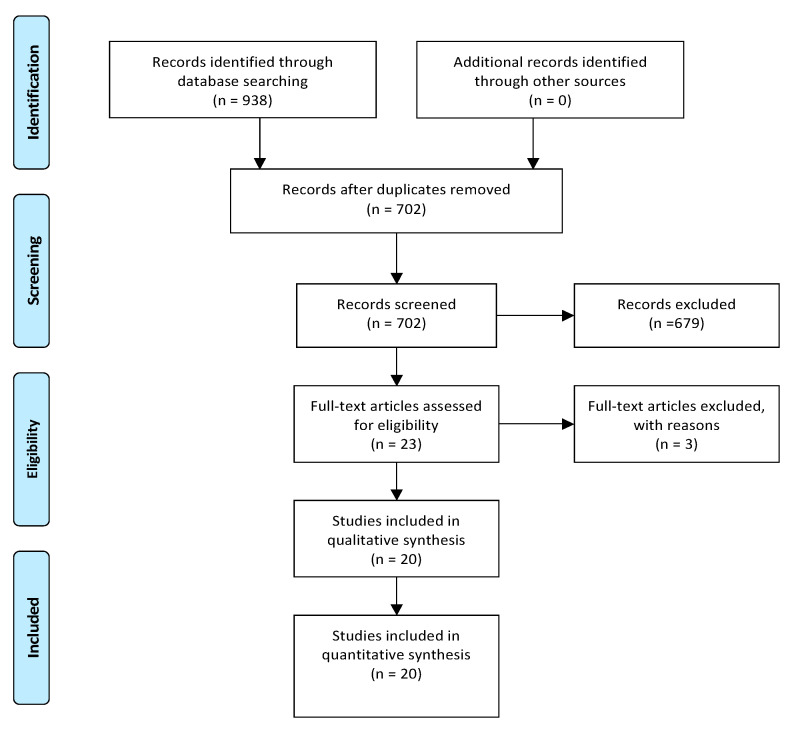
Search flow diagram.

**Table 1 diagnostics-11-00003-t001:** Pre-, intra-, and postoperative characteristics of the included patients.

Author; Year	QAS	Parity	Surgery for EM, Previous Surgery	Primary Symptom/Association with Menstruation	Type of Surgical Repair	Pathology-IC
Sun; 1979 [22]	5/5	NA	No	Painless swelling (7 year) became painful (4 year)/no	Hernia repair with mass excision and sac repair	Low grade papillary adenocarcinoma
Mesko; 1988 [23]	4/5	2	R inguinal herniorrhaphy, TAH-BSO	Gradually enlarging inguinal mass/no	Two inguinal cystic masses removal and exploratory laparotomy with RSO	Clear cell adenocarcinoma arising from NC- EM
Irvin; 1998 [24]	5/5	4	TAH-BSO (no EM-related)	Enlarging mass/no	Radical L hemivulvectomy with clitorectomy	Endometrial stromal sarcoma of the vulva, arising from the extrapelvic portion of the round ligament, within NC
Cervini; 2004 [25]	2/5	1	No	Gradually enlarging pubic mass/yes	Excisional biopsy	EM
Kirkpatrick; 2006 [26]	2/5	NA	NA	Painless mass-incidental finding in adrenal adenoma investigation/no	Biopsy (US guided)	EM
Wang; 2009 [27]	4/5	0	NA	Painful mass/yes	Laparoscopy for pelvic EM and excision of inguinal subcutaneous bulging mass	EM
Ito; 2010 [28]	3/5	NA	TAH-BSO (mucinous cystadenocarcinoma-stage Ia-14 years)	Mass/no	Tumor resection, mesh-plug repair	Endometrioid adenocarcinoma
Gaeta; 2010 [29]	3/5	NA	NA	4 inguinal painless masses; 2 inguinal discomfort; 2 cyclic pain	NA	NA
Albal; 2011 [30]	3/5	NA	No	Inguinal swelling/no	En block lump excision	Loculated cyst of NC with deposition of endometrial tissue
Bagul; 2011 [31]	3/5	NA	No	Painful mass/no	Aspiration of hydrocele fluid and sac transfixed and en bloc excised	Multiple foci of EM, hemosiderin deposition, inflammation and fibrosis
Jimnez; 2011 [32]	3/5	1	R nephrectomy	Painful mass/yes	Exploratory laparoscopy, cyst removal, and inguinal ring defect repair with mesh	NA
Bendon; 2011 [33]	4/5	0	IUD (prior 9 months), R inguinal hernia repair	Painful groin mass/yes	NC dissection and lesion within the sac excision	Fibroadipose tissue with evidence of EM
Noguchi; 2014 [34]	3/5	4	NA	Groin swelling; irregular genital bleeding; urine pregnancy test positive (hCG = 3090)	Exploratory laparotomy, extraction of right inguinal mass, closure of deep inguinal ring	Ectopic pregnancy and Nuck EM
Uno; 2014 [35]	4/5	0	NA	Painful groin mass/no	Inguinal approach, dissection, and cyst excision	Mesothelial cells lined the wall of the cyst with degeneration, inflammation, hemorrhage, formation of hyperplastic collagen fiber and hemosiderosis, IC: CD10(+), ER(+), PgR(+)
Okoshi; 2017 [36]	4/5	1	Laparoscopy for intrapelvic EM (10 years ago)	Pubic painful mass/no	Anterior open cyst excision and internal inguinal; ring repair with mesh	Cyst lined with mesothelial-like cells and accompanied by partial subcapsular hemorrhage, endometrium-like cystic wall IC: podoplanin & ER receptors
Motooka; 2018 [37]	5/5	2	No	Pubic painful and bleeding mass/no	Inguinal tumor resection, partial radical vulvectomy, clitoridectomy, R pectineal muscles, rectus abdominis muscles, oblique abdominal muscles, inguinal ligament and round ligament resection, bilateral inguinal LND-reconstruction with flaps	Endometrioid carcinoma associated with EM on the round ligament in R NC
Niitsu; 2019 [38] ^a^	5/5	NA	No	Inguinal mass /no (*n* = 1)	Open no hernia repair (*n* = 1)	EM
				Inguinal mass /no (*n* = 3)	Open, inguinal hernia repair (Marcy’s) (*n* = 3)	
				Inguinal mass/yes (*n* = 1)	Open, inguinal hernia repair (Marcy’s)-NC wall thickening (*n* = 3)	
				Inguinal mass/no (*n* = 2)		
				Inguinal mass/yes (*n* = 1)	Open, no inguinal hernia repair-NC wall thickening (*n* = 1)	
				Inguinal mass/no (*n* = 1)	Open no inguinal repair-NC wall thickening (*n* = 1)	
				Inguinal mass/no (*n* = 1)	Open hernia repair (mesh)-NC wall thickening (*n* = 1)	
2019; Raviraz [39]	3/5	0	No	Gradually enlarging, cyclically painful groin swelling, subfertility/yes	Open (patent processus vaginalis)	EM
Thomas; 2019 [40]	2/5	NA	No	Gradually enlarging cyclical painful groin swelling (1 year)/yes	Open excisional biopsy	Clusters of endometrial glands and stroma embedded in the fibrous tissue
Nagase; 2020 [41]	5/5	NA	No	Painful mass	Mass excision	Cyst lined by single-layered flat mesothelial cells- (hydrocele) with few glands of columnar cells accompanied by endometrial cells, hemorrhage and hemosiderin-laden macrophages, IC: single-layered flat cells calretinin +, podoplanin +, and WT-1 +, Columnar cells ER +, PR +, Round monotonous cells ER +, PR +, CD10 +, and WT-1 +

^a^ Case series (10 cases), QAS: quality assessment score, IC: immunohistochemistry, EM: endometriosis, R: right, L: left, TAH-BSO: total abdominal hysterectomy, bilateral salpingo-oophorectomy, LND: lymph node dissection, NA: not available, IDU: intrauterine device, NC: Nuck canal.

**Table 2 diagnostics-11-00003-t002:** Characteristics of studies reporting endometriosis associated Nuck canal malignancy.

Author; Year	Type of Surgery (after Diagnosis of Malignancy)	Histological Type	Recurrence
Motooka; 2018 [37]	Partial radical vulvectomy, clitoridectomy, & resection of the R pectineal muscles, rectus abdominis muscles, oblique abdominal muscles, inguinal ligament, and round ligament. Bilateral inguinal LND en bloc & EL	Well-differentiated endometrioid carcinoma, endometriosis- associated on the round ligament in the R NC Metastatic tumor was in inguinal and left external iliac lymph nodes	12 months after the 1st surgery endometrial cancer stage IA, Grade I and 5 months after 2nd endometrioid adenocarcinoma of the right ovary IA Grade I
Ito; 2010 [28]	Open resection of the tumor with mesh repair	Endometrioid adenocarcinoma of the NC	No
Irvin; 1998 [24]	Left radical hemivulvectomy and clitorectomy	Endometrial stromal sarcoma (ESS) of the vulva, arising from the extrapelvic portion of the round ligament, within the NC Tumor arose from EM	Lung metastasis 21 months after the 1st surgery (wedge resection of the metastatic pulmonary nodule) and 9 months later lung metastasis (right upper lobectomy)
Mesko; 1988 [23]	Mass excision, exploratory laparotomy, Right salpingo oophorectomy, right common iliac nodes sampling and bilateral inguinal LND after 3 months	Clear cell adenocarcinoma arising from Nuck EM	R groin tumor recurrence, multiple positive right iliac lymph nodes, one positive right obturator node, several positive right inguinal nodes (radiation therapy) and 2 years after first surgery, left pulmonary metastasis (refused chemo)
Sun; 1979 [22]	Excision with hernioplasty and EL	Low grade papillary adenocarcinoma	Local recurrence 3 years after first surgery resection of right lower anterior abdominal wall (inguinal ligament, canal and peritoneum) and groin dissection with exploratory laparotomy

LND: lymph node dissection; R: right; EL: exploratory laparotomy; NED: No evidence of disease, NC: Nuck canal, EM: endometriosis

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
