# Peer review of "Endometriosis of the Canal of Nuck: A Systematic Review of the Literature"

_diagnostics, 2020, doi:10.3390/diagnostics11010003_

Round 1

Reviewer 1 Report

This review is interesting but I think its design and presentation are not totally adequate.

  1. In the title and literature it appears as Canal of Nuck. However in the text, keywords and search strategy and data collection use the words: Nuck canal, Nuck cyst, Nuck endometriosis. It is right?
  2. Cases of inguinal endometriosis were not included. I do believe, however, that many cases have been reported as inguinal endometriosis. In Medline for "inguinal endometriosis" appear 1021 articles.
  3. See Basnayake O, et al (endometriosis of the inguinal canal ..) in Case Reports in Surgery, 2020, ID 8849317.
  4. The tables are too long and complex. I think all the data for a case should be on one line.
  5. The additional information in the tables is repeated in the text.
  6. From the point of view of the anatomy of the area and the origin of the round ligament, I would recommend that you consider the article “The female gubernaculum: role in the embryology and development of the genital tract and in the possible genesis of malformations” Europ J Obstet Gynecol Reprod Biol 2011, 159: 426-432, and the figures included there.

Author Response

Dear reviewer

We thank you for giving us the opportunity to revise our manuscript and for considering it for publication in your appreciated journal. Below you will find our answers to your comments. An extensive revision according to the recommendations of the reviewers has been performed. A manuscript with changes tracked has been attached.

We will be happy to address potential future questions.

Kind regards,

Prodromidou Anastasia MD

Reviewer(s)' Comments to Author with authors’ response:

Reviewer 1

1. In the title and literature it appears as Canal of Nuck. However in the text, keywords and search strategy and data collection use the words: Nuck canal, Nuck cyst, Nuck endometriosis. It is right?

Answer: We confirm that the referred terminology by the reviewer is used throughout the text with equivalent meaning. More specifically, the terms Canal of Nuck and Nuck canal present the same results in the search of the Pubmed so we can assume that they could be considered identical. Additionally, the term “Canal of Nuck” has been also added in the keywords section (page 1) as well as in the keywords stated in the materials and methods in the search strategy (page 3, line 96).

2. Cases of inguinal endometriosis were not included. I do believe, however, that many cases have been reported as inguinal endometriosis. In Medline for "inguinal endometriosis" appear 1021 articles.

Answer: As defined in the text, the canal of Nuck is the anatomical defect with persistent processus vaginalis in females due to incomplete or failure of obliteration of the protrusion of parietal peritoneum through the inguinal canal. The canal of Nuck could be considered a part of the inguinal canal. Therefore, the respective pathology of the canal including endometriosis could be considered a distinct entity of the overall pathology of the inguinal canal. We chose to focus on the nuck endometriosis and not including all cases of inguinal endometriosis. During our search we run through all the articles referred to inguinal pathology and selected the nuck endometriosis-associated.

3. See Basnayake O, et al (endometriosis of the inguinal canal ..) in Case Reports in Surgery, 2020, ID 8849317.

Answer: We would like to the reviewer for his constructive comment. However, our search was stopped in May 2020. We acknowledge that this is a constantly evolving filed, being the reason why we chose to review this distinct pathology, however, it is difficult to constantly update our search and manuscript on every review process. We believe our systematic review in its current form set a benchmark for new studies published every month. With regards to the proposed article by Basnayake et al, we have run through it with great interest and identify that the nuck pathology was considered in the differential diagnosis but inguinal endometriosis was the final diagnosis with no clearly reported Nuck canal involvement. Therefore, we could consider the study among the excluded ones.

4. The tables are too long and complex. I think all the data for a case should be on one line.

Answer: We agree with the reviewer. The tables are totally revised so as for the information reported in the text not to be depicted in tables. More specifically, Tables 1and 2 were fused to one table which includes only the information that could not be described in the text (pages 4-6). Some of them were extensive and thus for some studies inclusion in one line was not possible, despite the fact that the most summarized version was imprinted.

5. The additional information in the tables is repeated in the text.

Answer: The Tables were extensively revised as mentioned above.

6. From the point of view of the anatomy of the area and the origin of the round ligament, I would recommend that you consider the article “The female gubernaculum: role in the embryology and development of the genital tract and in the possible genesis of malformations” Europ J Obstet Gynecol Reprod Biol 2011, 159: 426-432, and the figures included there.

Answer: We would like to thank the reviewer for the comment and for proposing this really interesting article. We read carefully the article and added some additional information in the introduction section, describing the embryological origin of the round ligament and its association with the processus vaginalis in females which if persisted becomes the canal of nuck (page 2, lines 60-62). The respective references of the study by Acien et al has been added.

Reviewer 2 Report

1. Please review the figure n. 1 2. The tables are too complex and long. Can you simplify. The additional information in the tables is repeated in the text.

Author Response

Dear reviewer

We thank you for giving us the opportunity to revise our manuscript and for considering it for publication in your appreciated journal. Below you will find our answers to your comments. An extensive revision according to the recommendations of the reviewers has been performed. A manuscript with changes tracked has been attached.

We will be happy to address potential future questions.

Kind regards,

Prodromidou Anastasia MD

Reviewer 2

  1. Please review the figure n. 1 2. The tables are too complex and long. Can you simplify. The additional information in the tables is repeated in the text.

Answer: We would like to thank the reviewer for this comment. The tables were totally revised so as for the information reported in the text not to also depicted in the tables. More specifically, Tables 1and 2 were fused to one table which includes only the information that could not be described in the text (pages 4-6).

Reviewer 3 Report

This study focuses on endometriosis of the canal of Nuck, and reviewing the literature about surgically diagnosed endometriosis of the canal of Nuck.  Endometriosis of the canal of Nuck is rare disease, and its epidemiology is still unclear. 

Thus this report is meaningful to clarify this less common endometriosis, and they report important and interesting clinical features of it. 

One point should be discussed more; other types of inguinal endometriosis.  If we found inguinal endometriosis, endometriosis of the canal of Nuck, round ligament, and subcutaneous lesion are major differential diagnoses, and they are often confused.  This report focused on endometriosis of the canal of Nuck, and the cases which are not pathologically diagnosed are excluded from this study.  To avoid confusion with these endometriosis, disquisition about these differential diagnoses is needed in “1. Introduction” and “4. Discussion”.

Author Response

Dear reviewer

We thank you for giving us the opportunity to revise our manuscript and for considering it for publication in your appreciated journal. Below you will find our answers to your comments. An extensive revision according to the recommendations of the reviewers has been performed. A manuscript with changes tracked has been attached.

We will be happy to address potential future questions.

Kind regards,

Prodromidou Anastasia MD

Reviewer 3

This study focuses on endometriosis of the canal of Nuck, and reviewing the literature about surgically diagnosed endometriosis of the canal of Nuck.  Endometriosis of the canal of Nuck is rare disease, and its epidemiology is still unclear. 

Thus this report is meaningful to clarify this less common endometriosis, and they report important and interesting clinical features of it.  

One point should be discussed more; other types of inguinal endometriosis.  If we found inguinal endometriosis, endometriosis of the canal of Nuck, round ligament, and subcutaneous lesion are major differential diagnoses, and they are often confused.  This report focused on endometriosis of the canal of Nuck, and the cases which are not pathologically diagnosed are excluded from this study.  To avoid confusion with these endometriosis, disquisition about these differential diagnoses is needed in “1. Introduction” and “4. Discussion”.

Answer: We would like to thank the reviewer for the comments. Some further information has been added in the introduction section (page 2, lines 60-62 and lines 66-67) which further clarify the anatomy of the inguinal region, the anatomical relations of the structures of the groin region and point out the difficulty of the differential diagnosis. Additionally, in the discussion section page 3, 2nd paragraph (lines 250-252, 255-256 and 259-262), the differential diagnosis of a mass in the inguinal area is analysed along with the ways of identifying the structure from which inguinal endometriosis originates.

Round 2

Reviewer 1 Report

R2

The authors have conveniently revised this work and significantly improved it. For me it's ok